# microRNA-1 regulates sarcomere formation and suppresses smooth muscle gene expression in the mammalian heart

Amy Heidersbach[1,2,3], Chris Saxby[1], Karen Carver-Moore[1], Yu Huang[1], Yen-Sin Ang[1,2,3], Pieter J de Jong[4], Kathryn N Ivey[1,2]*, Deepak Srivastava[1,2,3]*

[1]Gladstone Institute of Cardiovascular Disease, San Francisco, United States; [2]Department of Pediatrics, University of California, San Francisco, San Francisco, United States; [3]Department of Biochemistry and Biophysics, University of California, San Francisco, San Francisco, United States; [4]Children's Hospital Oakland Research Institute, Oakland, United States

**Abstract** *microRNA-1 (miR-1)* is an evolutionarily conserved, striated muscle-enriched miRNA. Most mammalian genomes contain two copies of miR-1, and in mice, deletion of a single locus, *miR-1-2*, causes incompletely penetrant lethality and subtle cardiac defects. Here, we report that deletion of *miR-1-1* resulted in a phenotype similar to that of the *miR-1-2* mutant. Compound *miR-1* knockout mice died uniformly before weaning due to severe cardiac dysfunction. *miR-1*-null cardiomyocytes had abnormal sarcomere organization and decreased phosphorylation of the regulatory myosin light chain-2 (MLC2), a critical cytoskeletal regulator. The smooth muscle-restricted inhibitor of MLC2 phosphorylation, Telokin, was ectopically expressed in the myocardium, along with other smooth muscle genes. *miR-1* repressed Telokin expression through direct targeting and by repressing its transcriptional regulator, Myocardin. Our results reveal that *miR-1* is required for postnatal cardiac function and reinforces the striated muscle phenotype by regulating both transcriptional and effector nodes of the smooth muscle gene expression network.

*For correspondence: kivey@ gladstone.ucsf.edu (KNI); dsrivastava@gladstone.ucsf.edu (DS)

## Introduction

Cardiac gene expression is cooperatively regulated by an intertwined network of transcription factors and microRNAs (miRNAs) (*Srivastava, 2006*; *Cordes and Srivastava, 2009*). Perturbations in the activity or expression of factors within this network result in cardiac structural and functional defects in animal models and in humans. Among these, serum response factor (SRF) and Myocardin (Myocd) cooperate to directly regulate the myogenic gene program in both cardiac and smooth muscle (*Treisman, 1987*; *Wang et al., 2001*; *Chen et al., 2002*; *Sepulveda et al., 2002* reviewed in *Wang and Olson, 2004*). These factors transcriptionally regulate numerous miRNAs that, in turn, regulate transcription factors to reinforce specific cellular decisions and behavior. (*Kwon et al., 2005*; *Zhao et al., 2005*; *Niu et al., 2008*; *Cordes et al., 2009*).

miRNAs are small, ~21 nucleotide (nt), single-stranded RNAs that negatively regulate the stability and translation of mRNA transcripts. miRNAs target sequences within the 3' UTRs of mRNA transcripts that are highly complementary to the miRNA seed sequence (nt 2–8) and have imperfect complementarity outside of the seed region (*Valencia-Sanchez et al., 2006*). Due to the degenerate nature of miRNA:mRNA interactions, a single miRNA may have hundreds of mRNA targets (*Bartel, 2009*). Often miRNAs target multiple genes in a common pathway, thereby amplifying the effect of an individual miRNA on a given biological process (*Fish et al., 2008*; *Cordes et al., 2009*).

**eLife digest** MicroRNAs are tiny RNAs that do not encode proteins. Instead, they regulate the expression of genes by preventing protein-encoding messenger RNAs from being translated into protein. MicroRNAs are expressed throughout the body, including the heart, where the most abundant microRNA is called *miR-1*. This is encoded by two nearly identical genes: *miR-1-1* and *miR-1-2*.

Mice that lack the *miR-1-2* gene have various heart abnormalities, but generally survive because they still produce some *miR-1* from their remaining *miR-1-1* gene. Now, Heidersbach et al. have generated the first mice that specifically lack both *miR-1* genes, and shown that these animals die before weaning.

When viewed under the electron microscope, heart muscle from *miR-1* double knockout mice lacks the characteristic 'striped', or striated, appearance of normal heart muscle. Additionally, *miR-1* double knockout hearts have some gene expression characteristics more similar to the smooth muscle found in the gut and in the walls of blood vessels. Smooth muscle differs from striated muscle in that it lacks sarcomeres: these are bands of fibrous proteins, such as myosin, that are essential for muscle contraction.

In normal mice, an enzyme called MLCK contributes to the formation and function of sarcomeres by adding phosphate groups to myosin molecules. By contrast, in smooth muscle an enzyme called Telokin promotes phosphate group removal, and thus affects the function of sarcomeres. Heidersbach et al. showed that *miR-1* interacts directly with *Telokin* mRNA to prevent its expression in the heart, and simultaneously represses a protein called Myocardin, which directly activates transcription of *Telokin*. However, when *miR-1* is absent, as in the *miR-1* double knockout mice, *Telokin* is expressed in heart muscle, along with many other genes characteristic of smooth muscle.

As well as improving our understanding of the development and functioning of the heart, these findings should shed new light on the role of microRNAs in maintaining the patterns of gene expression that characterize unique cell fates.

*microRNA-1 (miR-1)* is a highly conserved miRNA and its expression is enriched specifically in cardiac and skeletal muscle. In mice, it is expressed in the heart and somites of the developing embryo during myogenic differentiation, beginning around embryonic day (E) 8.5 (**Zhao et al., 2005**; **Liu et al., 2007**). The cardiac expression of *miR-1* increases during development, with a dramatic rise in the post-natal period. RNA sequencing has revealed that *miR-1* is the most abundant miRNA in the adult mouse heart, representing up to 40% of all miRNA transcripts (**Rao et al., 2009**).

*miR-1* is transcribed as part of a bicistronic cluster with another striated muscle-enriched miRNA, *miR-133a*. In the genomes of most mammals, a duplication event has occurred resulting in two copies of the *miR-1/133a* locus, with *miR-1-2* and *miR-133a-1* on chromosome 18 and *miR-1-1* and *miR-133a-2* on chromosome 2 of the murine genome (**Figure 1—figure supplement 1**). Both precursors are transcriptionally regulated by several key myogenic transcription factors, including Myogenin, MYOD, SRF, MYOCD (**Kwon et al., 2005**; **Zhao et al., 2005**; **Rao, 2006**) and MEF2 (**Liu et al., 2007**). When processed, both the *miR-1-2* and *miR-1-1* precursors give rise to identical mature *miR-1* species, suggesting evolutionary pressure on both alleles (**Figure 1—figure supplement 2**). An additional miRNA cluster encoding *miR-133b* and *miR-206* is expressed uniquely in skeletal muscle, with the mature sequence of *miR-206* sharing a common seed with *miR-1*, but varying by 4 nts outside of the seed region.

Deletion of *miR-1-2* in mice (**Zhao et al., 2007**), which reduces the total expression of cardiac *miR-1* by roughly 50%, results in a spectrum of cardiac defects on a pure 129 background, including incompletely penetrant lethality, cardiomyocyte proliferative defects, and electrophysiological abnormalities. In flies, loss of the single *miR-1* gene results in abnormal myogenic differentiation and cell polarity defects in cardiac progenitors (**Kwon et al., 2005**; **Sokol, 2005**; **King et al., 2011**); however, the consequences of complete loss of *miR-1* in mammals are unknown.

In this study, we report that targeted deletion of the *miR-1-1* locus results in a phenotype similar to that described for *miR-1-2* null mice, and that the complete loss of *miR-1* is uniformly lethal before weaning due to cardiac dysfunction. We show that the loss of *miR-1* results in perinatal heart failure with myocardial sarcomeric defects, hypophosphorylation of Myosin Light Chain 2, and ectopic expression of Telokin, a smooth muscle-restricted inhibitor of Myosin Light Chain 2 phosphorylation. Furthermore,

we found the SRF co-factor, MYOCD, which is critical for transcriptional activation of both the cardiac and smooth muscle gene programs in vivo (*Li et al., 2003*; *Hoofnagle et al., 2011*; *Wang et al., 2001*; *Chen et al., 2002*), is directly targeted by *miR-1*. The smooth muscle isoform of *Myocd* was preferentially upregulated in the absence of *miR-1* and likely contributed to ectopic activation of the smooth muscle gene program in the heart. Our findings reveal that *miR-1* is embedded in an SRF-dependent cardiac gene program that promotes sarcomerogenesis and myogenic differentiation, while simultaneously repressing the smooth muscle program.

## Results

### Generation and characterization of *miR-1-1* null mice

We used homologous recombination to delete one allele of the *miR-1-1* precursor in embryonic stem (ES) cells, with a floxed neomycin cassette used for positive selection. (*Figure 1—figure supplement 3* and 'Materials and methods'). Injection of targeted ES cells into blastocysts resulted in high-percentage chimeras that transmitted the targeted allele through the germline. Intercrosses of *miR-1-1* heterozygous mice revealed that approximately half of all *miR-1-1* homozygous-null mice died before weaning when bred onto a pure 129 strain, similar to *miR-1-2* null mice (*Figure 1A*, upper). This lethality was strain dependent, as *miR-1-1* null animals on a mixed background (129/BL6) survived at normal Mendelian ratios until weaning. (*Figure 1A*, lower). By quantitative RT-PCR (qPCR), we found that total cardiac *miR-1* levels were decreased in *miR-1-1* knockout animals by about 40% at postnatal day (P) 2 (*Figure 1B*).

To evaluate cardiac function, we performed echocardiography on adult *miR-1-1* null or wild-type littermates. We found a reduction in fractional shortening, as well as an increase in left ventricular end-diastolic and end-systolic dimension in the *miR-1-1* knockout animals, indicating ventricular dilation (*Figure 1C*). Mild ventricular dilation was confirmed histologically (*Figure 1D I and II*). Additionally, we observed areas of fibrosis in the ventricular myocardium of the *miR-1-1* knockouts (*Figure 1D III and IV*). Like *miR-1-2* knockouts, *miR-1-1* knockout animals exhibited subtle conduction abnormalities, including prolonged ventricular depolarization and repolarization, indicated by a broader QRS complex and longer QT interval than controls. Additionally, *miR-1-1* knockout mice showed broader P waves without alteration of the PR interval (*Figure 1E*, *Figure 1—figure supplement 4*). Intermittent atrial arrhythmias were observed in two of five knockout animals analyzed, but were not observed in wild-type animals (*Figure 1—figure supplement 5*). Thus, mice lacking *miR-1-1* were grossly similar to those lacking *miR-1-2*, in that they exhibited partial lethality as well as subtle conduction abnormalities.

Cardiac conduction defects in the *miR-1-2* knockout mice were at least partially ascribed to dysregulation of the *miR-1* target, *Irx5*. Similarly, we found that *Irx5* was upregulated in the *miR-1-1* knockout mice (*Figure 1—figure supplement 6*). *miR-1-1* knockout mice also had a partial decrease in mature *miR-133a* levels (*Figure 1B*). Previously, using semi-quantitative RT-PCR, the *miR-133a* precursor levels were reported to be unchanged in the *miR-1-2* knockout animals. In this study, the analysis of the mature species by qPCR revealed a slight decrease in *miR-133a* in weaning-age *miR-1-2* null hearts, although not statistically significant (*Figure 1—figure supplement 7*). Importantly, *miR-133a* expression was maintained at a level described to be inconsequential in previous reports (*Liu et al., 2008*).

### Generation and characterization of compound *miR-1* knockout mice

To investigate the consequences of complete loss of *miR-1*, we intercrossed *miR-1-1* and *miR-1-2* mutant mice (*Zhao et al., 2007*) to generate double-heterozygous mice in a 129/BL6 mixed background. At weaning, no lethality was observed in the double-heterozygous mice (*Figure 2—figure supplement 1*) similar to the *miR-1-1* knockout on a mixed background. Using gene expression microarray analyses, we found 201 genes that were dysregulated in the single knockouts or double-heterozygous mice (*Figure 2—figure supplement 2*). Of those, 24 genes were coordinately dysregulated in animals of all three genotypes. The majority of genes (195/201) were similarly altered between the double heterozygotes and at least one of the single knockouts. There were, however, some differences in gene expression between these groups, which may suggest minor functional differences of the *miR-1* loci.

Double-heterozygous mice were subsequently intercrossed to generate knockout animals with only a single intact allele ($miR-1-1^{-/-}$: $miR-1-2^{+/-}$ and $miR-1-1^{+/-}$: $miR-1-2^{-/-}$). Most of these mice were viable and fertile on the mixed background, though mice lacking both the copies of *miR-1-2* were under-represented at weaning (*Figure 2—figure supplement 3*), indicating some difference in the compensatory ability of the two loci.

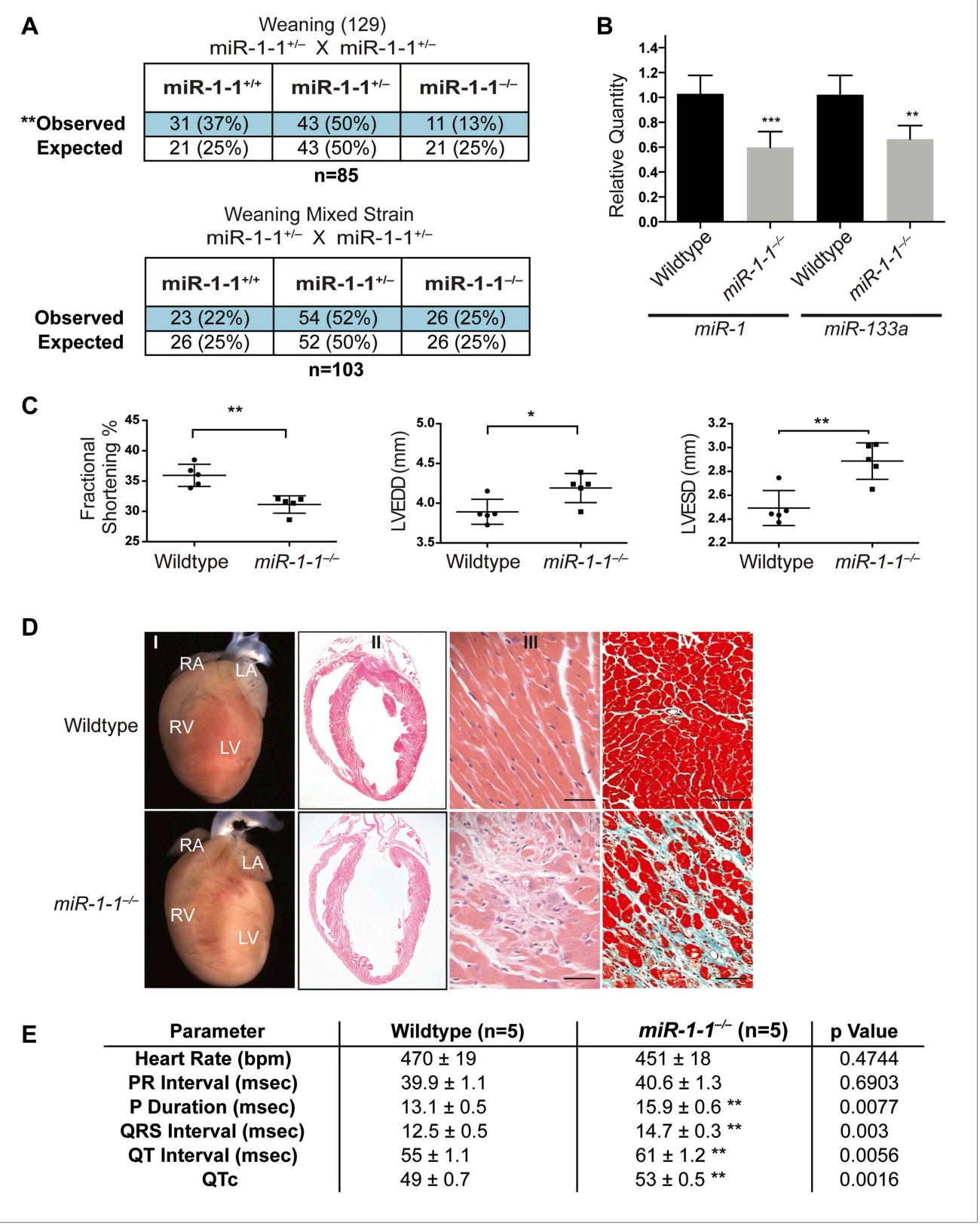

**Figure 1**. Viability and cardiac function of *miR-1-1⁻/⁻* mice. (**A**) Genotypes of offspring generated from *miR-1-1⁺/⁻* intercrosses on either a pure 129 background (upper) or a mixed BL6/129 strain (lower). Numbers of expected and observed genotype ratios are given for weaning-age (3-week-old) pups. (**B**) qPCR of mature *miR-1* and *miR-133a* in post-natal day 2 hearts. N = 6 per genotype. (**C**) Analyses of cardiac function by echocardiography of

*Figure 1. Continued on next page*

*Figure 1. Continued*

adult animals of indicated genotypes on a pure 129 background. N = 5 per genotype. LVEDD, left ventricular end-diastolic dimension; LVESD, left ventricular end-systolic dimension. (**D**) Adult wildtype (upper) and *miR-1-1⁻/⁻* (lower) hearts on a pure 129 background (I). RA, right atrium; LA, left atrium; RV, right ventricle; LV, left ventricle. Hematoxylin and eosin images taken at 1.25X magnification (II); and 40X magnification; scale bar indicates 25 µm (III). Masson trichrome stain of *miR-1-1* knockout myocardium, images taken at 40X; scale bar 50 µm (IV) (**E**) Analyses of cardiac conduction by electrocardiogram (EKG) of adult animals of indicated genotypes on a pure 129 background. *p<0.05; **p<0.01; ns, not significant.

The following figure supplements are available for figure 1:

**Figure supplement 1**. Schematic of the *miR-1/133a* genomic loci.

**Figure supplement 2**. Identical sequences of mature *miR-1-1* and *miR-1-2*.

**Figure supplement 3**. Left, targeting scheme for deletion of the *miR-1-1* locus.

**Figure supplement 4**. Averaged electrocardiogram tracings tracing from lead I of an adult wild-type or *miR-1-1* knockout animal on a pure 129 background.

**Figure supplement 5**. Electrocardiogram tracings of an adult wild-type or *miR-1-1* knockout animal on a pure 129 background.

**Figure supplement 6**. qPCR for the *miR-1* target, *Irx5,* in adult *miR-1-1⁻/⁻* and wild-type hearts.

**Figure supplement 7**. qPCR for mature *miR-133a* in adult wild-type and *miR-1-2* knockout hearts.

Single-allele mice (¾ alleles knocked-out) were intercrossed to generate mice completely lacking *miR-1*. *miR-1* compound-null mice on a mixed background were born at slightly less than Mendelian ratios and were of normal birth weight (***Figure 2A***, ***Figure 2—figure supplements 4 and 5***). Roughly a quarter of the double-knockout animals died very soon after birth. In a subset of these animals, we observed ventricular septal defects (VSDs) and misalignment of the aorta over the ventricular septum (overriding aorta), likely accounting for their lethality (***Figure 2—figure supplement 6***). Surviving *miR-1* double-knockouts failed to thrive post-natally, with no double-knockout animals surviving beyond P10 (***Figure 2B***, ***Figure 2—figure supplement 5***). Examination of surviving *miR-1* double-knockouts revealed expansion of the superior portion of the right ventricle (conus) and enlargement of the atria, when compared to wild-type mice beginning at P0 (***Figure 2C***, asterisk). By P4, and more extensively by P10, the dilation of all cardiac chambers was observed, with a particularly notable enlargement of the right atria. Echocardiography revealed severely impaired fractional shortening by P2 (***Figure 2D and E***) with poor systolic function. Frequent ventricular thrombi were observed by P4, consistent with impaired cardiac function in these animals (***Figure 2C***). Functional analysis of the conduction system by electrocardiogram (EKG) revealed a spectrum of abnormalities, including a prolonged QRS complex, and prolonged PR and QT intervals (***Figure 2F***, ***Figure 2—figure supplement 7***). The presence of frequent sinus pauses was observed in all knockout animals analyzed (***Figure 2—figure supplement 8***). These morphological, functional, and electro-physiological data indicate that the postnatal lethality in *miR-1* knockout animals is due to cardiac dysfunction.

qPCR confirmed that *miR-1* was not detectable in hearts of *miR-1* compound-null mice. Notably, *miR-133a* expression was decreased in *miR-1* double-knockout animals with the dysregulation of *miR-133a* becoming more pronounced with age as the locus normally becomes more actively transcribed (***Figure 2—figure supplement 9***). As *miR-1* lies transcriptionally upstream of *miR-133a*, we cannot exclude the possibility that the decrease in *miR-133a* expression is in part due to impaired transcriptional read-through downstream of the *miR-1* targeting event. However, chromatin immunoprecipitation revealed reduced RNA polymerase II (Pol II) occupancy at the *miR-1-1/miR-133a-2,* and *miR-1-2/miR-133a-1* promoters in P2 *miR-1* double-knockout mouse hearts, suggesting that transcriptional initiation of these loci is reduced secondary to loss of *miR-1* expression (***Figure 2—figure supplement 10***). This suggests a feedback mechanism whereby *miR-1* maintains appropriate expression of the *miR-1/133a* loci.

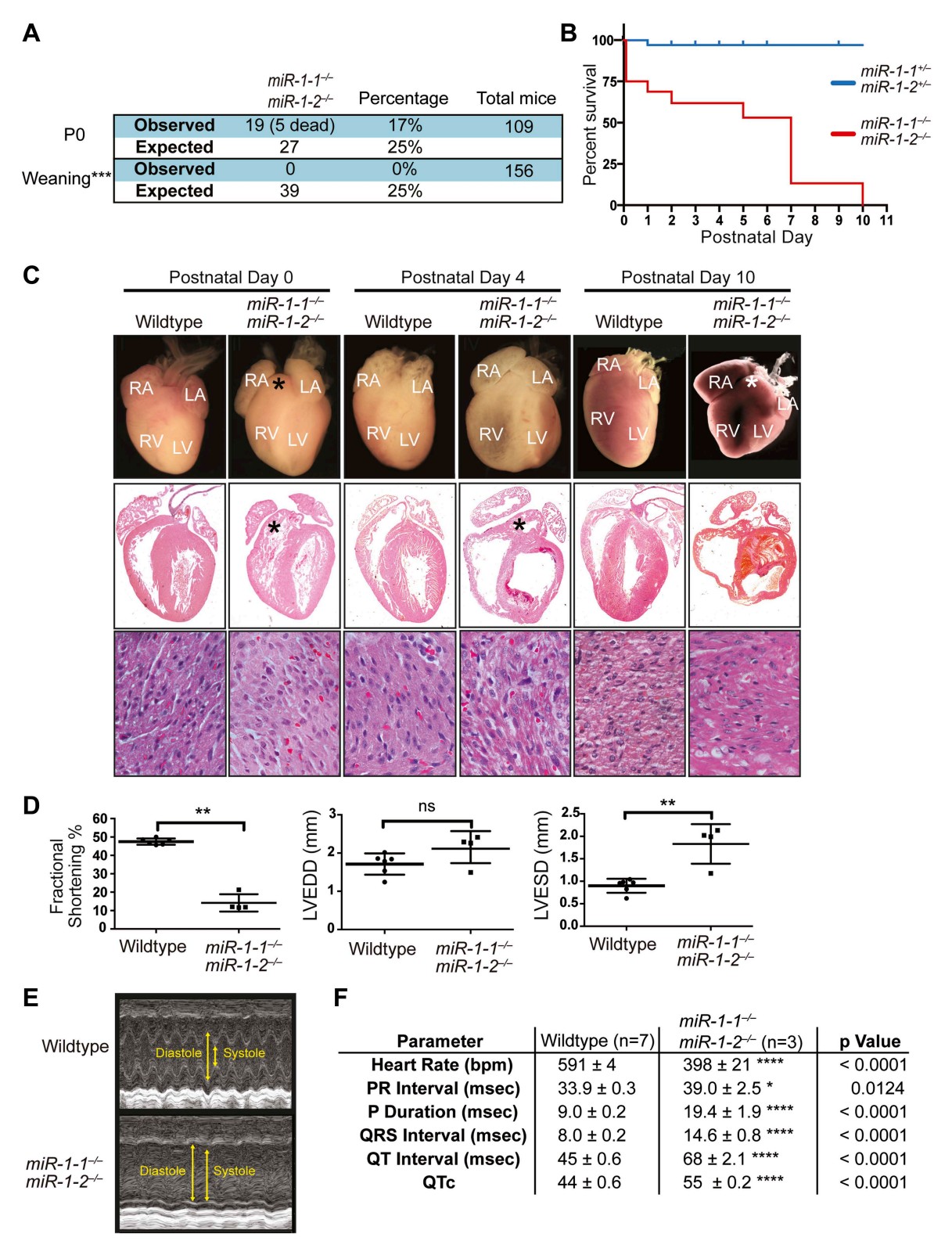

**Figure 2**. Compound *miR-1* knockout mice exhibit lethality due to a spectrum of cardiac defects. (**A**) Genotypes of offspring generated from
*miR-1-1⁻/⁺:miR-1-2⁻/⁻* X *miR-1-1⁻/⁻*: and *miR-1-2⁺/⁻* intercrosses on a mixed BL6/129 background. Numbers of expected and observed genotype
ratios are given for post-natal day 0 (P0) and weaning-age (3-week-old) pups. (**B**) Kaplan-Meier survival curve of *miR-1* double-knockout animals and
*Figure 2. Continued on next page*

*Figure 2. Continued*

double-heterozygous littermates. (**C**) Abnormal cardiac morphology in postnatal *miR-1* double-knockout mice includes an elongated outflow tract, as evidenced by bulging of the conus at P0 (asterisk). By P4, chamber dilation and thinning of the myocardium was apparent and ventricular clots were commonly observed. RA, right atrium; LA, left atrium; RV, right ventricle; LV, left ventricle. Middle panel images taken at 1.25X magnification; lower panel images taken at 40X magnification. (**D**) Echocardiography showed abnormal cardiac function in P2 *miR-1* null compared to wild-type animals. N = 7 for wild-type mice and N = 4 for *miR-1* null mice. LVEDD, left ventricular end-diastolic dimension; LVESD, left ventricular end-systolic dimension. (**E**) Representative M-mode image by echocardiography indicating diastolic and systolic dimensions. (**F**) Electrocardiographic analysis revealed conduction abnormalities in *miR-1* double-knockout animals by P2, including a decreased heart rate and elongated QRS relative to wild-type controls. *p<0.05; **p<0.01; ***p<0.001;****p<0.0001; ns, not significant.

The following figure supplements are available for figure 2:

**Figure supplement 1**. Genotypes of offspring generated from *miR-1-1+/−*: *miR-1-2+/+* X *miR-1-1+/+*: and *miR-1-2+/−* intercrosses on a mixed BL6/129 background.

**Figure supplement 2**. Cluster analysis of relative gene expression changes in *miR-1-1−/−* and *miR-1-2−/−* single knockout and *miR-1-1+/−:miR-1-2+/−* double-heterozygous hearts.

**Figure supplement 3**. Genotypes of offspring generated from *miR-1-1−/−*: *miR-1-2+/−* X *miR-1-1+/−*: *miR-1-2−/−* intercrosses on a mixed BL6/129 background at weaning.

**Figure supplement 4**. Genotypes of offspring generated from *miR-1-1−/−*: *miR-1-2+/−* X *miR-1-1+/−*; and *miR-1-2−/−* intercrosses on a mixed BL6/129 background at birth.

**Figure supplement 5**. *miR-1* double-knockouts fail to thrive.

**Figure supplement 6**. Abnormal cardiac morphology in *miR-1* double-knockouts found dead at birth compared to wild-type control.

**Figure supplement 7**. Averaged electrocardiogram tracings from lead I of P2 *miR-1* null (right) or wild-type (left) mice.

**Figure supplement 8**. Electrocardiogram tracings of a postnatal wild-type or *miR-1* null animal on a mixed background.

**Figure supplement 9**. *Left*, qPCR of mature *miR-1* or *miR-133a* in E12.5 wild-type or *miR-1* null hearts (n = 3 per genotype). *Right*, mature miRNA expression in P0 wild-type or *miR-1* null hearts (n = 5 per genotype).

**Figure supplement 10**. qPCR to detect the *miR-1-2/133a-1* or *miR-1-1/133a-2* promoter sequences, or an intergenic genomic sequence, following chromatin immunoprecipitation (ChIP) of RNA polymerase II (RNA Pol II) in P2 wild-type or *miR-1* null hearts.

## Cardiomyocytes lacking *miR-1* show sarcomeric defects

Given the relatively normal cardiac morphogenesis in surviving *miR-1* knockouts, we hypothesized that their impaired cardiac contractility was due to a primary myocardial defect. In agreement, transmission electron microscopy of ventricular tissue revealed the areas of extensive sarcomeric disruption at P0, before the onset of ventricular dilation (***Figure 3A***). Mitochondrial morphology was also abnormal, including overall decreased mitochondrial size (***Figure 3—figure supplements 1 and 2***) and decreased complexity of mitochondrial cristae (***Figure 3—figure supplement 1***).

Sarcomeric morphology was further analyzed in isolated P0 neonatal cardiomyocytes by immunostaining for the Z-line protein α-Actinin, as well as by Phalloidin staining to visualize the filamentous actin cytoskeleton. Individual cardiomyocytes were classified on a scale from I–V based on their sarcomeric organization, with class-I cells showing highly ordered sarcomeres, and class-V cells having no sarcomeric organization (***Figure 3—figure supplement 3***). Sarcomeres in *miR-1* null cardiomyocytes showed significant disruption when compared to those from *miR-1* double-heterozygous littermates. Only 10% of double-knockout cardiomyocytes had highly organized sarcomeres, compared to 45% of *miR-1* double-heterozygous cells. The degree of sarcomeric organization correlated with *miR-1* dosage as cardiomyocytes lacking three out of four copies of *miR-1* showed an intermediate degree of organization (***Figure 3C***).

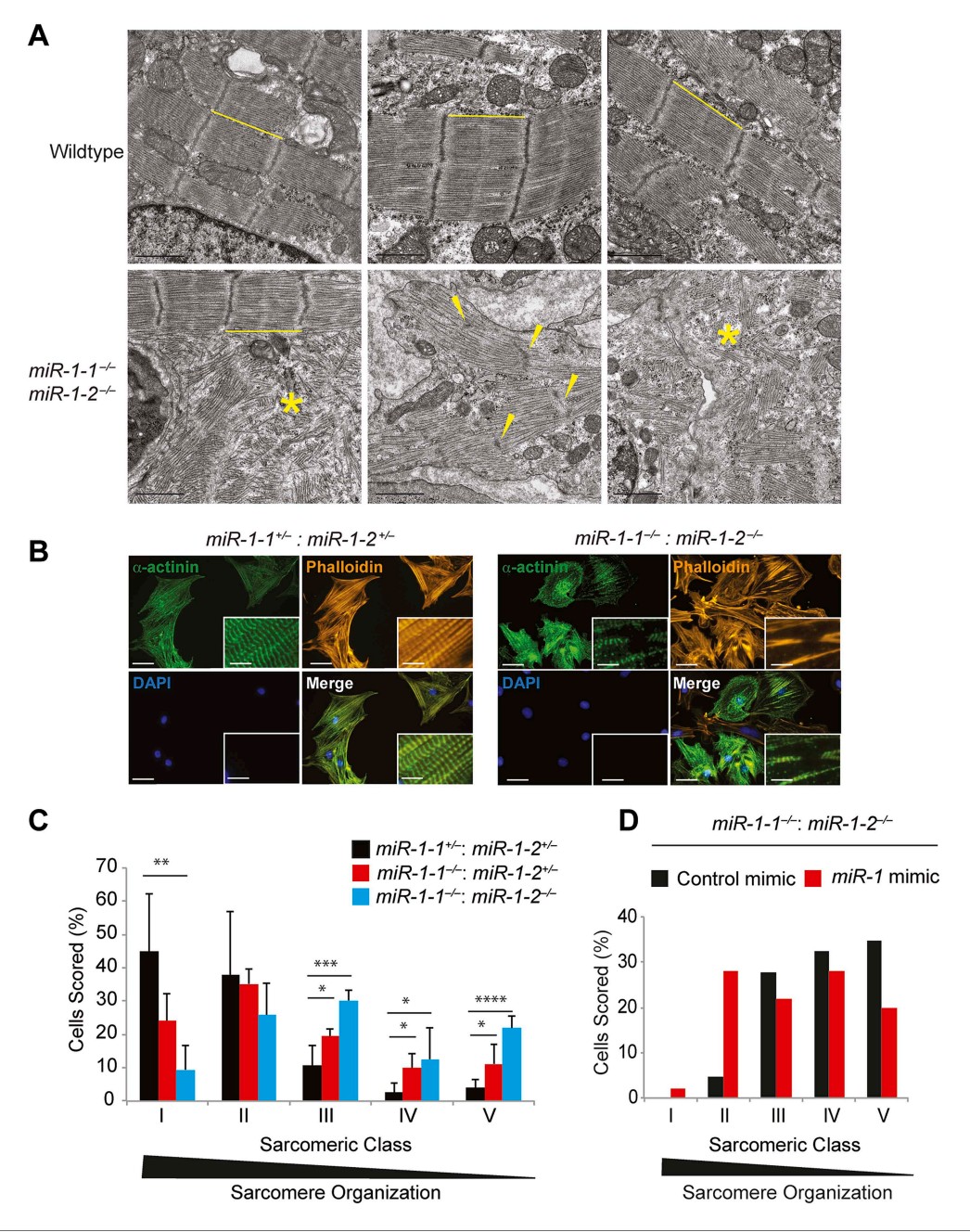

**Figure 3**. Sarcomere disruption in *miR-1* null cardiomyocytes. (**A**) Transmission electron microscopy (TEM) of P0 wild-type or *miR-1* double-knockout myocardium. Representative areas of sarcomeric disarray are indicated (*). Distance between Z-lines is indicated with lines. Arrows indicate disrupted Z-line structures. Scale bar, 1 μm. (**B**) Immunofluorescence of sarcomeric structures in isolated P0 cardiomyocytes with Phalloidin (F-actin cytoskeleton [orange]) and sarcomeric alpha-actinin (green) DAPI (blue) indicates nuclei. Images captured at 40X magnification. (**C**) Percentage of cardiomyocytes of individual sarcomeric classes observed. N = 3 for *miR-1* null mice; N = 2 for *miR-1-1*$^{-/-}$:*miR-1-2*$^{+/-}$ mice; and n = 6 for *miR-1* double-heterozygous mice, with a minimum of 50 cells classified per animal. (**D**) Representative analysis showing percentage of P5 *miR-1* null cardiomyocytes of individual sarcomeric classes observed 24 hr after transfection of a *miR-1* mimic or control RNA mimic. Roughly 50 cardiomyocytes per condition were evaluated. *p<0.05; **p<0.01; ***p<0.001;****p<0.0001; ns, not significant.

The following figure supplements are available for figure 3:

*Figure 3. Continued on next page*

*Figure 3. Continued*

**Figure supplement 1**. Transmission electron microscopy (TEM) reveals mitochondrial morphology defects in *miR-1* null hearts.

**Figure supplement 2**. Quantification of mitochondrial area from TEM images reveals a reduction in mitochondrial area in both *miR-1* double-knockouts compared to controls, although the degree to which mitochondrial area is reduced varies.

**Figure supplement 3**. Sarcomeric organization classification scheme.

To determine if reintroduction of *miR-1* was sufficient to improve sarcomere organization in cardiomyocytes from post-natal hearts, we transiently transfected a *miR-1* RNA mimic into cultured cardiomyocytes isolated from P5 animals. Indeed, adding *miR-1* partially rescued this phenotype and enhanced sarcomeric organization in *miR-1* double-knockout cardiomyocytes, compared to those treated with a control RNA mimic (*Figure 3D*).

## Dysregulation of *Telokin* and myosin light chain phosphorylation

We performed RNA sequencing of late embryonic stage (E18) *miR-1* null and wild-type hearts to identify genes that were dysregulated in the absence of *miR-1*. We selected this time point in order to reveal primary changes in gene expression due to the loss of *miR-1* and not those that may arise secondary to heart failure. Given that many direct miRNA targets are upregulated at the protein, but not transcript level, we expected that sequencing analysis of this stage would identify pathways that are dysregulated in the *miR-1* knockout, some of which may involve direct *miR-1* targets. We utilized the GREAT interface (*McLean et al., 2010*) to evaluate the enrichment of miRNA targets within the set of genes that were upregulated in the *miR-1* null hearts, compared to wild-type controls (*Figure 4— figure supplement 1*). We found that genes containing *miR-1/206* seed sequence complementarity were most significantly enriched in this data set. Genes targeted by *miR-495*, *miR-518a-2*, *miR-501* and *miR-409* were also enriched, though to a lesser degree. Notably, mRNAs with seed sequence complementarity to *miR-133a* were not enriched, suggesting that the reduction in *miR-133a* levels did not reach the threshold for significant dysregulation of genes in the *miR-1* knockout.

We next performed a gene ontology analysis using the GO-Elite interface (*Zambon et al., 2012*) to determine at a functional level which dysregulated genes may be phenotypically relevant. We found that genes participating in metabolic and mitochondria-related pathways were downregulated in *miR-1* knockout, compared to wild-type hearts, consistent with the morphological abnormalities visualized by electron microscopy (*Figure 4A*). Interestingly, many upregulated genes fell into the 'regulation of actin cytoskeletal' pathway (*Figure 4B*). As the dysregulation of actins and other cytoskeletal genes contribute to cardiomyopathies, in a complementary analysis, we investigated if any of the upregulated cytoskeletal genes were direct *miR-1* targets. miRNA targets remain challenging to predict computationally; therefore, we utilized three different target prediction algorithms—Targetscan (www.targetscan.org), PITA (http://genie.weizmann.ac.il/pubs/mir07/mir07_dyn_data.html), and Pictar (http://pictar.mdc-berlin.de/cgi-bin/PicTar_vertebrate.cgi)—to identify consensus predicted targets (*Figure 4C*, *Figure 4— source data 1*). Of the genes that were expressed at an equal or higher level in the knockout compared to wild-type hearts, 89 genes were predicted *miR-1* targets by all three algorithms. Of these, 13 genes were significantly upregulated at the mRNA level in the *miR-1* null hearts. (*Figure 4C*, *Figure 4— source data 1*).

In the *miR-1* null animals, myosin light chain kinase (MLCK) was a putative *miR-1* target of particular interest, as it was highly upregulated and had a known role in regulating the cytoskeleton (*Figure 4B,C*). MLCK, which is encoded within the *Mylk* locus, regulates the cytoskeleton by phosphorylating the regulatory Myosin Light Chain 2 (MLC2). MLC2 is associated with Myosin Heavy Chain (MHC) and is situated adjacent to the actin interaction domain of the globular myosin head. In smooth muscle, MLC2 phosphorylation is sufficient to induce contraction, and in the striated muscle of the heart, phosphorylation increases the rate and magnitude of contractile force (*Davis et al., 2001*, reviewed in *Kamm and Stull (2001)*).

The 31 exons of the *Mylk* locus give rise to four distinct but overlapping gene products (*Figure 5A*). Two 220-kD MLCK isoforms, which vary only by alternate inclusion of exon 1, are expressed in non-muscle

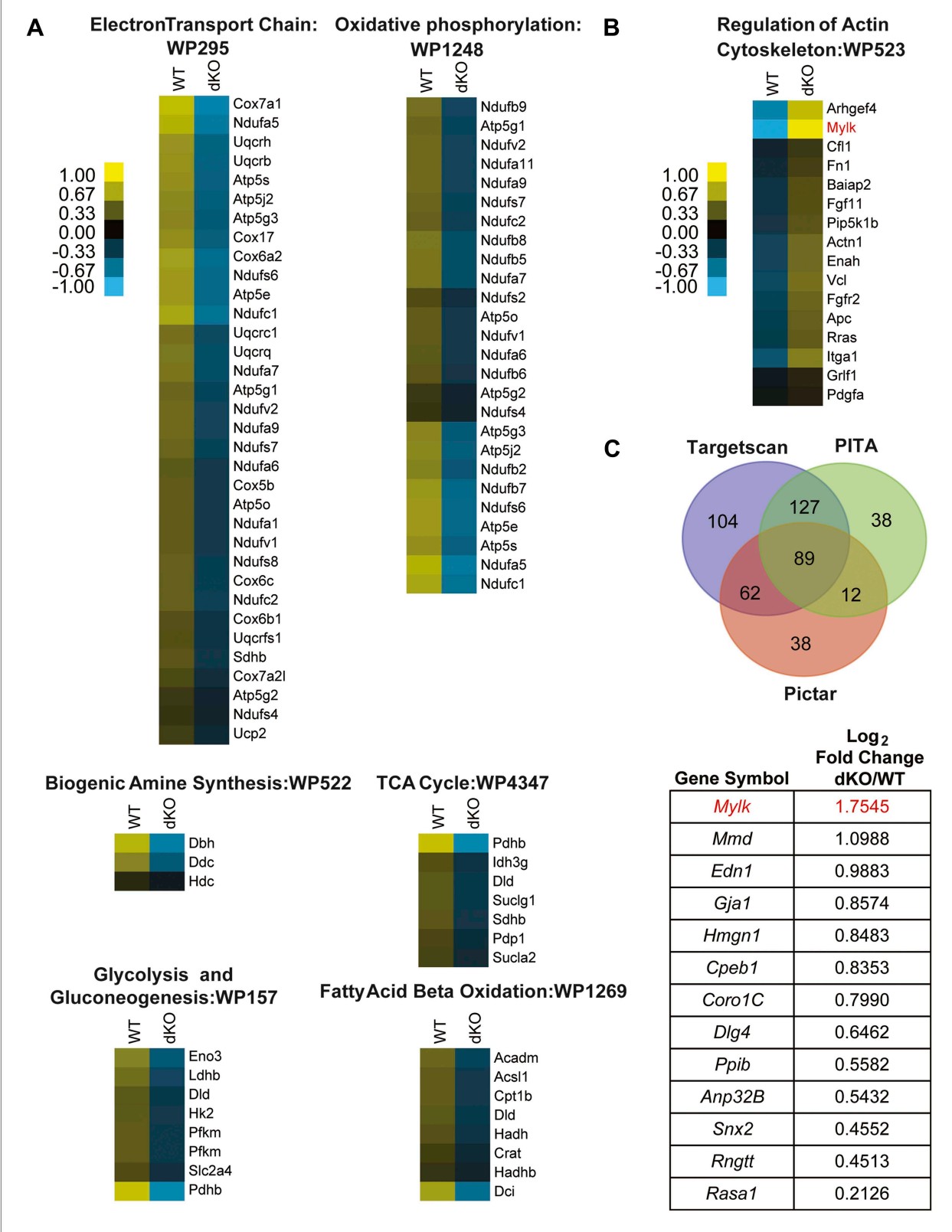

**Figure 4**. Pathway analysis of genes dysregulated in E18 *miR-1* null hearts. (**A**) Pathway analysis of genes downregulated in E18 *miR-1* null hearts identified via Go-elite and GenMAPP. Pathways related to metabolism are enriched. (**B**) Pathway analysis of genes upregulated in the *miR-1* null hearts identified via Go-elite and GenMAPP. Values represent normalized mean centered log$_2$ of FPKM for each genotype. (**C**) Venn diagram depicting overlap
*Figure 4. Continued on next page*

*Figure 4. Continued*

of *miR-1* targets as predicted by three prediction algorithms: Targetscan, PITA and Pictar. The genes analyzed were expressed at a relative quantity of ≥1 in *miR-1* null vs wild-type hearts based on their FPKM (upper). Consensus targets predicted by all three algorithms, showing significant upregulation in the *miR-1* null vs wild-type hearts are presented as descending $\log_2$ fold change in *miR-1* null over wild-type hearts (lower). *Mylk (MLCK)* in red was identified as a regulator of the actin cytoskeleton (**B**) and a significantly upregulated, predicted *miR-1* target (**C**). dKO, double-knockout; WT, wild-type; FPKM, fragments per kilobase per million.

The following source data and figure supplements are available for figure 4:

**Source data 1**. *miR-1* targets as predicted by three algorithms.

**Figure supplement 1**. MicroRNA target enrichment analysis of genes that were expressed at a relative quantity of ≥1 in the *miR-1* null vs wild-type hearts.

cell types, a 130-kD broadly expressed isoform and a 17-kD isoform, called Telokin, which lacks a functional kinase domain but shares an identical carboxy-terminal domain, and is specifically expressed in smooth, but not cardiac, muscle. The transcription of each isoform is regulated by individual promoters, which direct their spatially and temporally restricted expression (*Herring et al., 2006*). When the 3′ UTR region of *Mlck*/*Telokin* containing the predicted *miR-1* binding site was cloned downstream of a luciferase reporter, luciferase activity was repressed in the presence of a *miR-1* mimic. The repression was alleviated when the target site was deleted, validating this 3′ UTR of *Mlck/Telokin* as a direct *miR-1* target (*Figure 5B*).

To determine which isoforms were upregulated in *miR-1* double-knockout hearts, we examined the protein expression profiles of the various isoforms in wild-type and *miR-1* double-knockout hearts by western blot (*Figure 5C*). The non-muscle 210-kD isoform was undetectable, and the expression of the 130-kD MLCK isoform was not significantly altered in *miR-1* double-knockout hearts. Telokin protein, as previously reported (*Herring et al., 2001*), was not detectable in wild-type hearts, but was highly expressed in hearts of *miR-1* double-knockouts. qPCR confirmed the normally smooth muscle-restricted *Telokin* was aberrantly expressed in *miR-1* null myocardium (*Figure 5C*). Notably, while smooth muscle gene expression is reported to be dysregulated in *miR-133a-1*:*miR-133a-2* double-knockout hearts (*Liu et al., 2008*)**,** we found that *Telokin* expression was not upregulated in those animals (*Figure 5—figure supplement 1*), suggesting that the misexpression was specifically due to the loss of *miR-1*.

Despite lacking a catalytic kinase domain, Telokin plays an important role in the regulation of MLC phosphorylation in smooth muscle by inhibiting MLCK and promoting activity of the MLC phosphatase (*Choudhury et al., 2004*; *Khromov et al., 2012*) (*Figure 5D*). Consistent with aberrant Telokin expression in cardiomyocytes, MLC phosphorylation was dramatically decreased in *miR-1* double-knockout hearts, likely contributing to the observed cardiac dysfunction (*Figure 5E*).

While direct repression by *miR-1* may help to inhibit cardiac translation and stability of *Telokin* transcripts, it was unclear how loss of *miR-1* resulted in the preferential upregulation of *Telokin* and not the full-length *Mlck*, which is thought to share a common 3′ UTR. To determine if the *Telokin* promoter was aberrantly active in *miR-1* double-knockout hearts, we performed RNA Pol II chromatin immunoprecipitation and assayed for occupancy at the *Telokin* promoter. In wild-type hearts, Pol II occupancy at the *Telokin* promoter was equivalent to that of an untranscribed intergenic region, in agreement with *Telokin's* published smooth muscle-restricted expression pattern (*Herring et al., 2001*) (*Figure 5F*). In contrast, Pol II actively bound the *Telokin* promoter in *miR-1* double-knockout hearts (*Figure 5F*). These data indicate that *miR-1* normally acts to repress *Telokin* expression in the heart by both directly targeting the *Telokin* 3′ UTR and by negatively regulating *Telokin* transcription.

## *miR-1* targets the SRF co-factor Myocardin for repression

To gain mechanistic insight into how *miR-1* may be negatively regulating *Telokin* transcription and to identify, at a more general level, transcriptional networks perturbed in the absence of *miR-1*, we again utilized the GREAT interface (*McLean et al., 2010*) to identify known transcription factor motifs within the Msig database that were enriched within regulatory elements of the genes dysregulated in the double-knockouts. Interestingly, a disproportionate number of the genes upregulated in *miR-1* double-knockout hearts contained CArG boxes (CC/ATn/GG), a motif bound by SRF (*Treisman, 1986*) (*Figure 6A*, *Figure 6—figure supplement 1*). SRF is a critical regulator of the cardiac and smooth muscle transcriptome and regulates sarcomere formation in cardiomyocytes (*Li et al., 1997*, *2003*; *Wang et al.,*

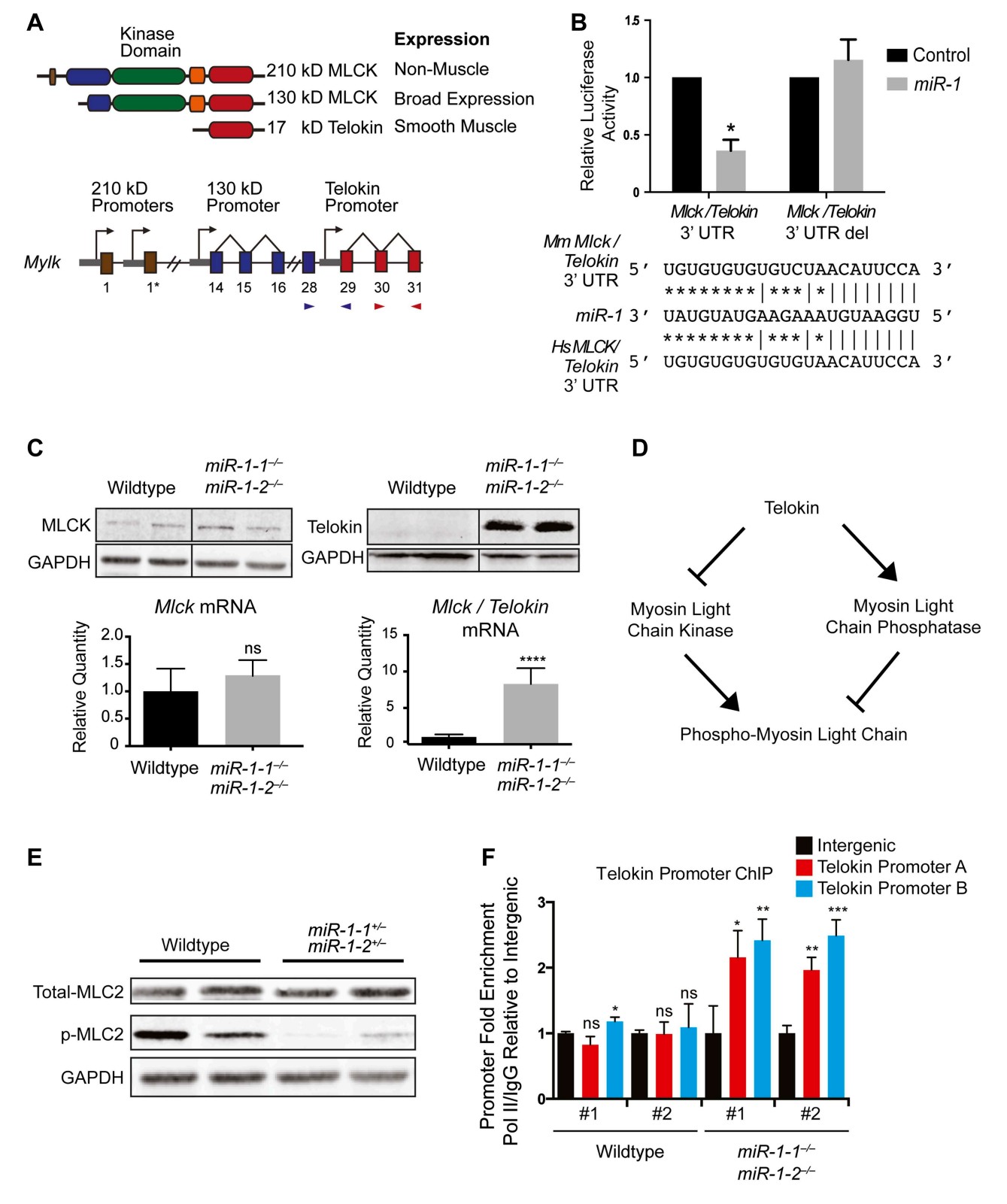

**Figure 5**. Dysregulation of Telokin and Myosin light chain phosphorylation in *miR-1* null hearts. (**A**) Diagram of the gene products encoded within the *Mylk* locus. Independent promoters preceding exons are indicated by arrows. Exon-spanning qPCR primers used are indicated in red (*Mlck/Telokin*) or blue (*Mlck*). (**B**) Luciferase activity of a reporter construct containing ~200 bp of the *Mlck/Telokin* 3′-UTR surrounding the predicted *miR-1* binding site
*Figure 5. Continued on next page*

*Figure 5. Continued*

with or without the site deleted. The constructs were co-transfected into H9C2 myoblasts with a *miR-1* mimic or a control mimic. Sequence of the putative *miR-1* target site as predicted by Targetscan and site conservation between human (Hs) and mouse (Mm) is indicated. (**C**) Western blot of heart lysates (top) and qPCR of RNA (bottom) from P0 wild-type or *miR-1* null mice. (N = 5 per group). (**D**) Model of Telokin function in smooth muscle to promote the activity of myosin light chain phosphatase and inhibit the activity of the myosin light chain kinase. (**E**) Western blot of total myosin light chain 2 (MLC2) and phosphorylated myosin light chain (p-MLC2) in P0 wild-type or *miR-1* null hearts; GAPDH serves as loading control. (**F**) qPCR of the *Telokin* promoter sequence or an intergenic genomic sequence following chromatin immunoprecipitation (ChIP) of RNA polymerase II in P2 wild-type or *miR-1* null hearts. For the *Telokin* promoter, two non-overlapping probe sets were used, indicated as *Telokin* promoter A and B.

The following figure supplements are available for figure 5:

**Figure supplement 1**. Putative *miR-1* targets dysregulated in *miR-1* null hearts were not affected in *miR-133a* double-knockout hearts.

---

*2001*; *Niu et al., 2007*; *Hoofnagle et al., 2011*). SRF is also a highly conserved direct upstream regulator of the *miR-1/133a* transcript (*Kwon et al., 2005*; *Zhao et al., 2005*) and is itself a target of *miR-133a*, being upregulated in the hearts of *miR-133a-1:miR-133a-2* double-knockout animals (*Chen et al., 2005*; *Liu et al., 2008*). However, SRF was not dysregulated at the RNA or protein level in *miR-1* double-knockout hearts (*Figure 6B*). This finding suggests that a simple increase in SRF expression is not responsible for the upregulation of SRF target genes and, furthermore, that the level of *miR-133a* expressed in *miR-1* double-knockouts is sufficient to maintain normal SRF expression levels.

One mechanism by which the specificity of SRF targets is conferred is through the expression and activity of tissue-specific cofactors. Myocardin (MYOCD), which is expressed predominantly in cardiac and smooth muscle (*Wang et al., 2001*), is one such cofactor. MYOCD is necessary and sufficient for smooth muscle differentiation in vitro and in vivo (*Li et al., 2003*; *Chen et al., 2002*; *Wang et al., 2003*) and is necessary for the differentiation and maintenance of ventricular cardiomyocytes (*Wang et al., 2001*; *Hoofnagle et al., 2011*). Among the SRF target genes that were dysregulated in the *miR-1* double-knockouts, many, including *Telokin*, were known targets of the MYOCD/SRF complex (*Figure 6C*, *Figure 6—figure supplement 1*), while MYOCD independent SRF target genes, such as *Fos* and *Egr1*, were not upregulated (*Figure 6C*). Notably, the expression of numerous other known SRF co-factors, including NKX2-5, MRTF-A, BRG1, GATA4 and GATA6 were unchanged (*Figure 6—figure supplement 2*). These findings suggest that MYOCD/SRF dependent gene expression is specifically upregulated in the *miR-1* knockout hearts.

Using qPCR to compare the expression of *Myocd* in *miR-1* double-knockout hearts and controls, we found that the expression of this critical transcription factor was increased roughly twofold upon deletion of *miR-1* (*Figure 7A*). Using Targetscan, we identified a potential *miR-1* binding site in the *Myocd* 3'-UTR and demonstrated, using a luciferase assay, that *miR-1* directly targets the *Myocd* 3' UTR (*Figure 7B*). Previous studies have shown that the *Telokin* promoter is more responsive to MYOCD than the full-length 130-kD *Mlck* promoter (*Herring et al., 2006*). Thus, as we observed, an increase in MYOCD expression would be predicted to preferentially upregulate the transcription of the *Telokin* isoform, while only modestly increasing transcription of the full-length transcript.

## Smooth muscle *Myocd* expression is upregulated in *miR-1* double-knockout hearts

There are two distinct *Myocd* isoforms, the full-length cardiac isoform (cMYOCD) and the truncated smooth muscle isoform (smMYOCD) (*Creemers et al., 2006*). *smMyocd* is produced by alternate splicing and inclusion of exon2a, which encodes a translational stop and necessitates the use of an alternate downstream start site. This splicing event results in a truncated protein, lacking the N-terminal–MEF2 interaction domain of the full-length cMYOCD. Although the functional significance of these two isoforms is not well understood, the promoters of some smooth muscle-restricted genes, such as *Tagln* (*Sm22*), are more sensitive to smMYOCD than cMYOCD, while some cardiac promoters, such as *Myh6* (*α-Mhc*), are more responsive to cMYOCD (*Imamura et al., 2010*). Normally, *smMyocd* expression is relatively low in the neonatal heart; however, we discovered using qPCR that in *miR-1* double-knockout hearts the expression of *smMyocd* was fivefold higher than control hearts, while *cMyocd* was upregulated roughly 1.8-fold (*Figure 7A*). More specific upregulation of *smMyocd* was not evident with statistical confidence in our RNA-seq data likely due to insufficient sequencing depth to detect the small 44 bp exon 2a inclusion within the relatively low overall abundance of *smMyocd* transcript.

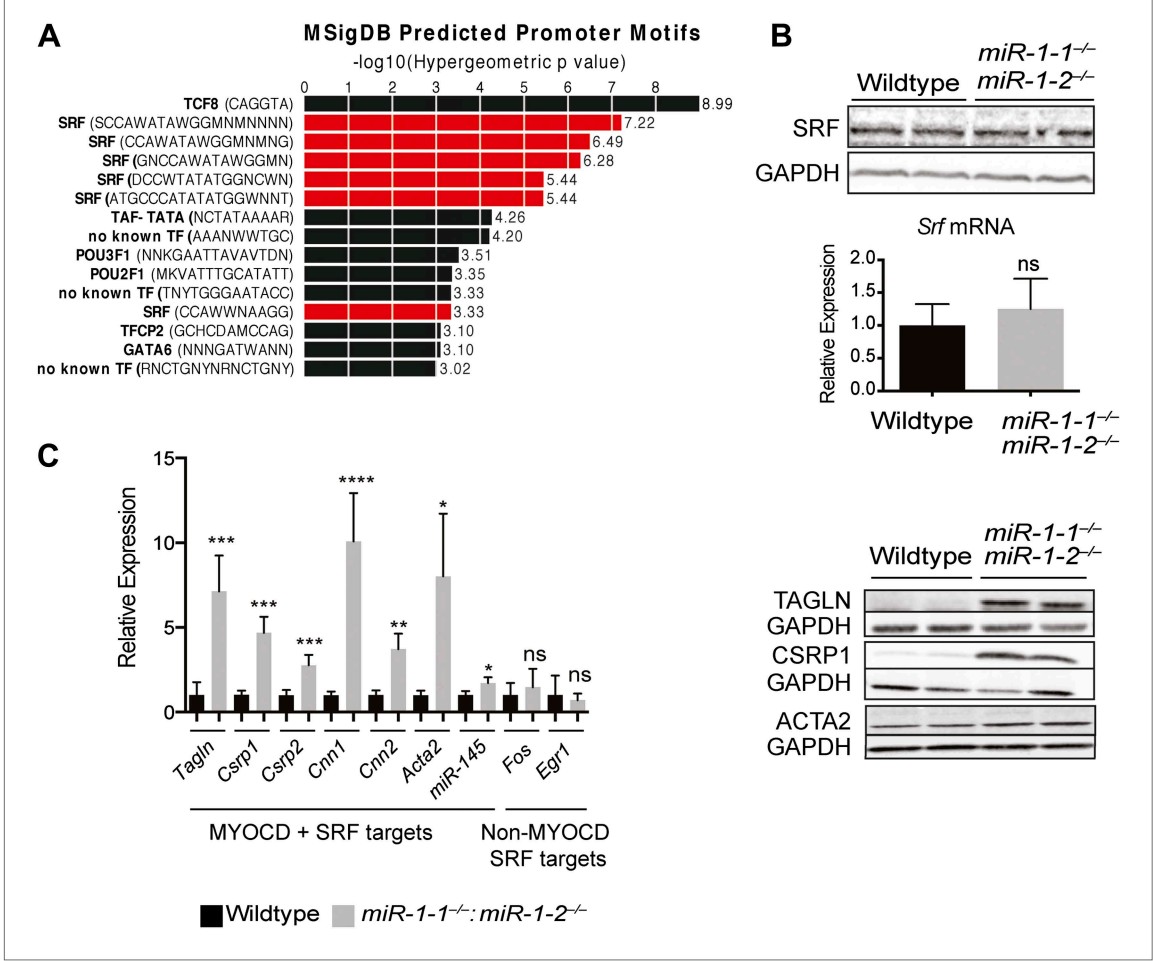

**Figure 6**. Upregulation of SRF targets in *miR-1* null hearts. (**A**) Promoter motif enrichment in genes upregulated in *miR-1* double-knockout hearts. Motif sequences and transcriptional regulators are indicated (left) with the $\log_{10}$ of the hypergeometric p value graphed. Multiple CArG box sequences, the motif bound by SRF, were identified (red). (**B**) Western blot (top), and qPCR (bottom) for *Srf* expression in post-natal *miR-1* null or wild-type hearts. (**C**) Myocardin dependent and independent SRF target gene expression in postnatal *miR-1* null or wild-type hearts by qPCR (left). Western blots of selected Myocardin-dependent SRF target genes (right). *Tagln, Transgelin/Sm22; Csrp1,2, Cysteine And Glycine-Rich Protein 1,2; Cnn1,2, Calponin1,2; Acta2, smooth muscle alpha actin; miR-145, microRNA-145; Fos, FBJ Murine Osteosarcoma Viral Oncogene Homolog; Egr1, Early growth response 1.* *p<0.05; **p<0.01; ***p<0.001;****p<0.0001; ns, not significant.

The following figure supplements are available for figure 6:

**Figure supplement 1**. Dysregulation of SRF targets in *miR-1* double-knockout hearts.

**Figure supplement 2**. (A) qPCR of SRF co-factors in *miR-1* wild-type or double-knockout hearts. N = 5. (B) Western blot for protein expression of SRF co-factors. ns, not significant.

While c*Myocd* still represents the predominant isoform in *miR-1* double-knockout hearts (data not shown), subtle changes in transcription factor expression can result in dramatic changes in gene expression. Using a luciferase reporter cloned downstream of the mouse *Telokin* promoter, we found that smMYOCD was twice as effective as cMYOCD at activating the *Telokin* promoter (***Figure 7C***). More broadly, we found that many of the MYOCD-dependent genes upregulated in the *miR-1* knockouts were smooth muscle genes (***Figure 6—figure supplement 1***). Smooth muscle gene upregulation was also reported in the myocardium of *miR-133a* knockout mice, due in part to the upregulation of SRF. Neither total *Myocd* nor sm*Myocd,* however, was upregulated in the myocardium of *miR-133a* double-knockouts, indicating that the *miR-133a* levels do not affect *Myocd* expression (***Figure 5—figure supplement 1***). These data suggest a model whereby *miR-1 and miR-133a cooperate to repress*

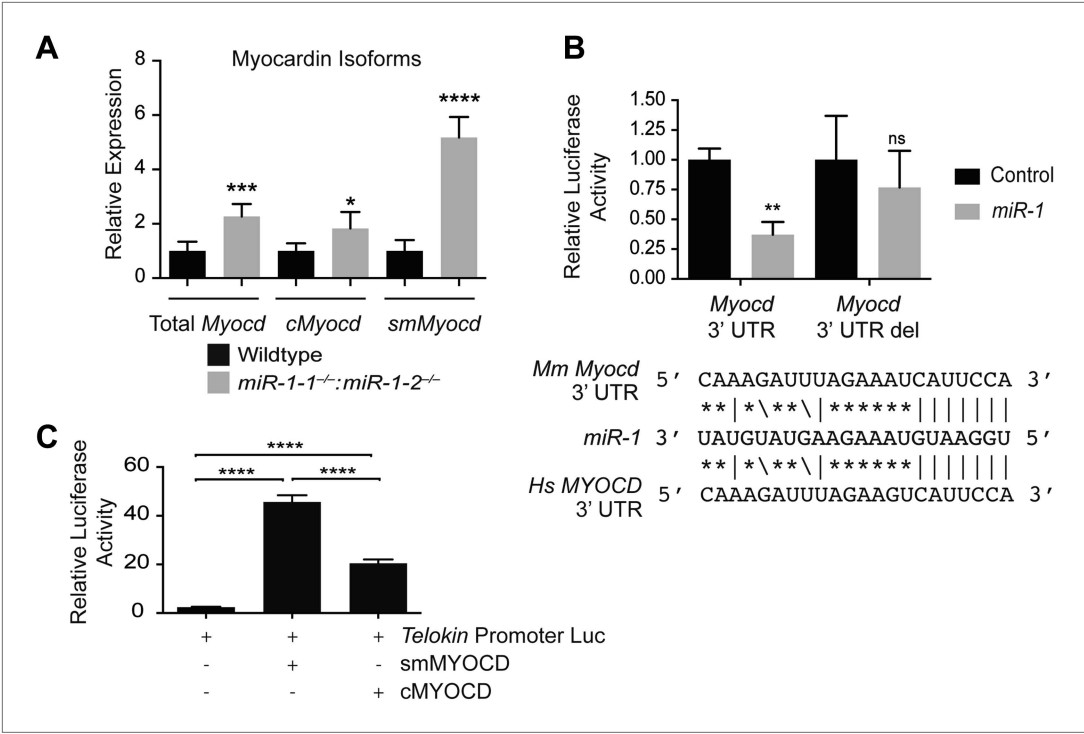

**Figure 7**. *miR-1* regulates the SRF co-factor, Myocardin. (**A**) qPCR for *Myocardin (Myocd)* expression. Probe sets are specific for inclusion of exon 2a for *smooth muscle Myocd (smMyocd)*, exclusion of exon 2a for *cardiac Myocd (cMyocd)*, or for downstream exons common to both transcripts (Total *Myocd*). P0 hearts were analyzed, N = 5 per genotype. (**B**) Luciferase activity of a reporter construct containing the *miR-1* putative target site derived from the *Myocd* 3' UTR or a deleted site. The constructs were co-transfected into H9C2 myoblasts with either a *miR-1* mimic or a control mimic. The sequence of the putative *miR-1* target site as predicted by Targetscan and site conservation between human (Hs) and mouse (Mm) is indicated. (**C**) The expression of luciferase driven by the *Telokin+370* promoter when transfected into Cos cells, alone or with either full-length *smMyocd* or *cMyocd*. *p<0.05; **p<0.01; ***p<0.001;****p<0.0001; ns, not significant.

smooth muscle gene transcription in the heart by repressing *smMyocd* and *Srf*, respectively, thereby reinforcing the striated muscle phenotype (*Figure 8B*).

## Discussion

The results of this study reveal the essential, multi-faceted role that *miR-1* plays in the mammalian heart. The uniform lethality of *miR-1* double-knockout animals, associated with dilated cardiomyopathy and abnormal heart rhythm, indicates that *miR-1* is required for normal cardiac contractility. At a cellular level, we found that sarcomere disruption is one feature that underlies the impaired cardiac function, and reintroduction of *miR-1* into ex vivo cardiomyocytes was sufficient to partially rescue this phenotype. *miR-1* functions to negatively regulate the smooth muscle-specific inhibitor of MLC phosphorylation, Telokin. Aberrant upregulation of Telokin and the resulting decrease in phosphorylated MLC in *miR-1* null hearts may contribute, in part, to the sarcomeric contractility defect. *miR-1* regulation of *Telokin* is accomplished by direct targeting of its 3' UTR, as well as through targeting of its transcriptional regulator, *Myocd*. In particular, *miR-1* preferentially regulates *smMyocd*, thereby complementing the effects of *miR-133* to suppress the smooth muscle gene program (*Figure 8*).

### *miR-1* is required for normal sarcomere formation and maintenance

In ex vivo cardiomyocytes, restoring *miR-1* expression alone was sufficient to partially rescue the sarcomeric defects of *miR-1* double-knockout hearts, demonstrating that the loss of *miR-1* plays a causative role in this phenotype, but not ruling out a contribution from decreased *miR-133a* expression. In this and previous studies (*Mishima et al., 2009*), genes related to the actin cytoskeleton are highly responsive to alterations in *miR-1* expression. MLC phosphorylation is critical for sarcomere assembly

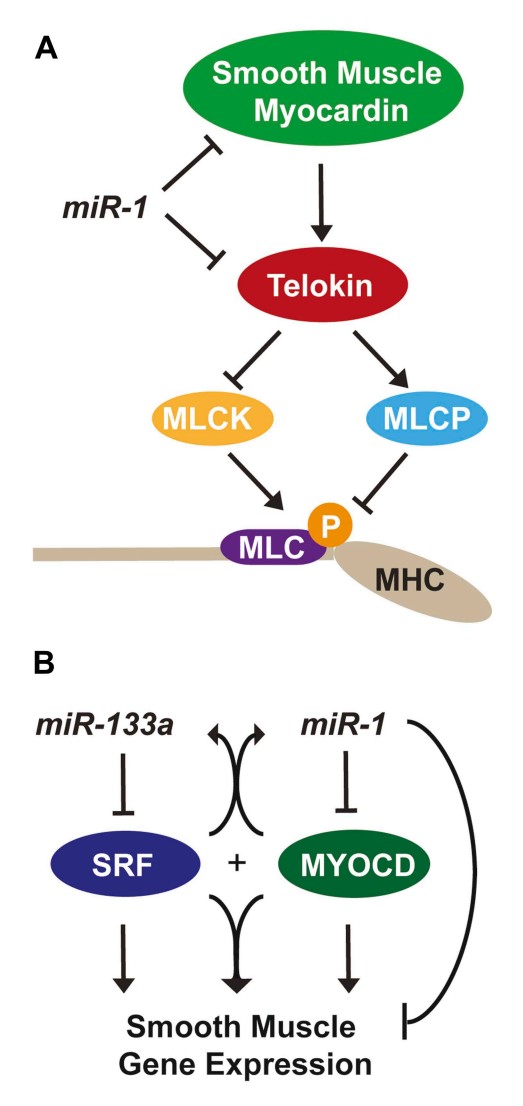

**Figure 8**. *miR-1* regulation of Myosin light chain phosphorylation and smooth muscle gene expression. (**A**) Dual regulatory model by which *miR-1* acts to normally repress *Telokin* expression in the heart by directly targeting its 3′ UTR and its upstream transcriptional regulator, smooth muscle myocarding (smMYOCD). MLCK, myosin light chain kinase; MLCP, myosin light chain phosphatase; MLC, myosin light chain; MHC, myosin heavy chain; P, phosphorylation. (**B**) Regulatory model by which the *miR-1/133a* cluster cooperates to suppress smooth muscle gene expression in the heart. DOI: 10.7554/eLife.01323.035

and regulating the speed and force of contraction in the heart (*Aoki et al., 2000*; *Davis et al., 2001*). In mammals, there are multiple MLC2 isoforms, but deletion of the single cardiac zebrafish isoform results in a lack of thick filament assembly (*Rottbauer et al., 2006*). At a molecular level, phosphorylation of MLC2 results in a conformational shift in the associated MHC, bringing it into closer proximity to the thin filament and increasing the probability of cross-bridge formation. Not surprisingly, in humans, *MLC2* mutations are associated with myopathies (*Poetter et al., 1996*; *Davis et al., 2001*). Thus, decreased phosphorylation of MLC in the absence of *miR-1* likely plays a role in the sarcomeric disruption and cardiac dysfunction in *miR-1* double-knockouts.

We found that one mechanism by which *miR-1* normally acts to maintain p-MLC in the heart is through the repression of the usually smooth muscle-restricted protein Telokin. Telokin expression in smooth muscle is thought to maintain cells in a less contractile state, and it is interesting that the heart employs *miR-1* to ensure the absence of Telokin through both transcriptional and post-transcriptional mechanisms (*Figure 8A*). Although it is not possible to distinguish the contribution of transcriptional regulation of Telokin vs direct *miR-1* targeting of the *Telokin* 3′ UTR, the dramatic degree of Telokin protein upregulation compared to the more moderate *Telokin* mRNA increase is consistent with some degree of translational control by *miR-1* (*Figure 4C*). How Telokin misexpression affects cardiac function has not been directly evaluated; however, ectopic expression in the myocardium has been reported in a doxorubicin induced rat model of cardiomyopathy, although the precise mechanism by which Telokin contributes to this pathology is unknown (*Dudnakova et al., 2003*).

While the focus of this study was to explore the function of *miR-1* in the heart, *miR-1* likely plays a more subtle role in the skeletal muscle. It is worth noting that electron microscopy of P0 *miR-1*-null skeletal muscle showed the presence of ordered sarcomeric structures (data not shown) and neonatal mice were mobile. While we cannot rule out a more subtle skeletal muscle phenotype due to the early lethality, it is possible that the skeletal muscle specific *miR-1* family member, *miR-206*, can compensate for the loss of *miR-1* in this tissue. A clear understanding of the role that the *miR-1* family plays in the context of skeletal muscle will require the generation a skeletal muscle-specific compound deletion of *miR-1-1*, *miR-1-2* and *miR-206*.

## *miR-133a* downregulation in *miR-1* null hearts

The marked downregulation of *miR-133a* in the *miR-1* double-knockouts is a critical consideration with respect to the interpretation of some of the phenotypes that we observed. While targeted disruption of *miR-1* might adversely affect transcriptional read-through of the *miR-1-1/miR-133a-2* locus,

secondary downregulation of *miR-133a* expression from regulatory feedback mechanisms also appears to occur (*Figure 2—figure supplement 10*). *miR-1* and *miR-133a* are dramatically upregulated during the differentiation of cardiac and skeletal muscle (*Chen et al., 2005*; *Ivey et al., 2008*). While the current study shows definitively that *miR-1* is not required for initial cardiomyocyte differentiation in vivo, the loss of *miR-1* expression may impair further cardiomyocyte differentiation and/or maturation. This would predictably result in decreased developmental upregulation of the *miR-1/133* loci. Consistent with this model, the loosely organized sarcomeric morphology and the abnormal mitochondrial morphology we observed by TEM are similar to those of more immature cardiomyocytes (*Hom et al., 2011*). The upregulation of members of the cardiac fetal gene expression program, such as *Nppa* (*Figure 6—figure supplement 1*), and even the upregulation of smooth muscle genes, which are normally expressed in embryonic cardiomyocytes, supports this paradigm. As such, the expression of *miR-133a*, a key developmentally regulated miRNA, may be secondarily downregulated in *miR-1* double-knockout mice as a result of a subtle cardiac differentiation or maturation defect, although the exact mechanism is unknown.

## The *miR-1* and *miR-133a* clusters cooperate to normally repress smooth muscle gene expression in the heart

The power of miRNAs to affect biological processes is amplified by the ability of a single miRNA to regulate multiple nodes within a genetic network. *Srf* is a direct target of *miR-133a*, and its expression, as well as that of many smooth muscle target genes, is upregulated in the hearts of *miR-133a*-knockout animals (*Chen et al., 2005*; *Liu et al., 2008*). We found that SRF mRNA and protein expression in *miR-1* double-knockout hearts was unaffected, despite the partial downregulation of *miR-133a*. Nevertheless, some of the smooth muscle genes upregulated in *miR-133a* double-knockout hearts were also upregulated in the *miR-1* double-knockout animals (*Acta2, Cald1, Csrp2, Cnn1, Tagln),* while others, including *Telokin, cMyocd* and *smMyocd*, were uniquely upregulated in the *miR-1* double-knockouts. Our data suggest that *miR-1* and *miR-133a*, transcribed together from bicistronic miRNA clusters, cooperate to repress SRF-dependent smooth muscle gene expression in the heart by independently regulating SRF or its co-factor, MYOCD, respectively (*Figure 8B*).

We also found that the *Telokin* promoter is preferentially responsive to the smMYOCD isoform, upregulated in the absence of *miR-1*. This is consistent with the observation that, while both MYOCD isoforms share an identical SRF-interacting domain, they are not functionally equivalent with respect to mediating the transcription of SRF target genes (*Creemers et al., 2006*; *Imamura et al., 2010*). While yet to be carefully evaluated, *cMyocd* and *smMyocd* are thought to share a common 3′ UTR; thus, it is unlikely that direct targeting by *miR-1* alone is responsible for the observed isoform switch. The splice factor(s) that regulate the inclusion or exclusion of the 2a exon in the smooth muscle or heart, respectively, are unknown. Therefore, a potential mechanism by which *miR-1* may regulate *Myocd* splicing is through the direct repression of a splice factor that normally mediates alternate inclusion of the 2a exon in smooth muscle. Further studies will be required to elucidate the mechanism by which *miR-1* regulates this critical transcription factor and ultimately smooth muscle gene expression.

## Materials and methods

### *miR-1-1* gene targeting and mouse breeding

The *miR-1-1* targeting vector was generated by flanking 5′ and 3′ genomic DNA around a floxed neomycin resistance gene (*Neo*) driven by a pGK promoter. Targeting of the *miR-1-1* locus was accomplished through homologous recombination in E14 ES cells and resulted in replacement of the genomic sequence containing the *pre-miR-1-1* (~280 bp) by the *Neo* gene. 672 colonies were screened, and three targeted clones were identified. *miR-1-1* heterozygous embryonic stem cells were injected into D3.5 BL6 blastocysts. For the establishment of the *miR-1-1*-targeted allele on a 129 background, chimeric males were mated to wild-type 129 (Jackson Lab) females. To generate compound *miR-1* knockouts, *miR-1-1*-targeted animals were crossed to a previously described *miR-1-2* targeted mouse maintained on a mixed (129/BL6) background. Animals double heterozygous for both *miR-1* alleles (*miR-1-1⁺/⁻:miR-1-2⁺/⁻*) were subsequently intercrossed to generate animals lacking three out of four alleles of *miR-1* (*miR-1-1⁻/⁻:miR-1-2⁺/⁻* or *miR-1-1⁺/⁻:miR-1-2⁻/⁻).* These animals were subsequently intercrossed to generate compound *miR-1* knockout animals (*miR-1-1⁻/⁻:miR-1-2⁻/⁻*). Genotyping was

performed by PCR [*Supplementary file 1*] with primers that specifically recognized the wild-type or targeted *miR-1-1* or *miR-1-2* alleles.

The animals were sacrificed by decapitation for neonates or by $CO_2$, followed by cervical dislocation for adult animals. All animal care and experimental protocols were reviewed and approved by the Institutional Animal Care and Use Committee of the University of California, San Francisco (UCSF).

## Electron microscopy

For electron microscopy, tissue was fixed in 2% glutaraldehyde, 1% paraformaldehyde in 0.1 M sodium cacodylate buffer pH 7.4, post fixed in 2% osmium tetroxide in the same buffer, en block stained in 2% aqueous uranyl acetate, dehydrated in acetone, infiltrated, and embedded in LX-112 resin (Ladd Research Industries, Williston, VT). Toluidine blue stained semi-thin sections were made to locate the areas of interest. The samples were ultrathin sectioned on a Reichert Ultracut S ultramicrotome and counter stained with 0.8% lead citrate. Grids were examined on a JEOL JEM-1230 transmission electron microscope (JEOL USA, Inc., Pleasanton, CA) and photographed with the Gatan Ultrascan 1000 digital camera (Gatan Inc., Pleasanton, CA).

## Neonatal transthoracic echocardiography and electrocardiogram

Adult mouse echocardiography was performed under anesthesia as described (*Qian et al., 2011*). Electrocardiograms were performed as described (*Zhao et al., 2007*). Neonatal studies were performed as described above without anesthesia.

## Histology and immunocytochemistry

Tissue was collected at indicated times and fixed using 10% formalin overnight at 4°C and stored subsequently in 70% ethanol. Paraffin embedding and staining was performed using standard histological techniques. Sarcomeric staining was performed using anti-sarcomeric alpha-actinin (1:400; Sigma, St. Louis, MO) and rhodamine conjugated Phalloidin (Clontech, Mountain View, CA).

## Isolation and culture of primary neonatal cardiomyocytes

Hearts were isolated from P0 animals and rinsed several times in 1X HBSS with Pen-Strep. The great vessels and atrium were removed and discarded, and the ventricles were minced manually with scissors and further disassociated enzymatically with collagenase digestion. 1 mg/ml collagenase II was added to the minced ventricles and briefly incubated at 37°C for 3–5 min with agitation. The digests were allowed to settle without agitation briefly (3 min) before the supernatant of this initial digestion (mostly blood cells) was discarded and another digestion was performed for 15–20 min. The supernatant of this subsequent digestion (cardiomyocytes) was added to 3X volume of FBS, and centrifuged for 5 min at 300 × *g* in a tabletop centrifuge. The resulting cell pellet was resuspended in DMEM/F12 medium with 10% FBS and Pen-Strep and passed through a 70-μM filter before plating on 1% gelatin with fibronectin. Cardiomyocytes for sarcomeric analysis were fixed and stained 24 hr post-plating. The cardiomyocytes used for sarcomeric rescue studies were transfected with *miR-1* mimic or control (Ambion/Life Technologies, Carlsbad, CA) roughly 12 hr post-plating with Lipofectamine 2000 (Life Technologies) and maintained in serum-containing medium. The cells were fixed for analysis 24 hr post-transfection.

## Cloning, plasmids, transfection and luciferase assays

Roughly 200 bp surrounding the predicted *miR-1* target sites in the *Mlck/Telokin* or *Myocd* 3' UTR were amplified directly from cDNA generated from *miR-1* double-knockout hearts with the primers listed (*Supplementary file 1*) and subcloned into the PGL3 (Promega, Madison, WI) firefly luciferase vector 3' of the reporter gene with XbaI.

Luciferase constructs were transfected along with a Renilla normalization vector into the H9C2 rat myoblast cell line with Lipofectamine 2000 (Life Technologies), according to manufacturer's instructions. Briefly, 12-well plates were transfected at roughly 60% confluency and analyzed 20 hr later. Each well received 3 μl of Lipofectamine 2000, 800 ng of PGL3-Target, and 200 ng of Renilla vector. Experimental wells received 10 pmols of *miR-1* mimic (Ambion/Life Technologies), while control wells received 10 pmols of a non-targeting control mimic (Ambion/Life Technologies). Promoter activity assays were performed with the *Telokin*+370 promoter (a gift from Dr Paul Herring, Indiana University). c*Myocd* and sm*Myocd* constructs were as described in *Wang and Olson (2004) Cordes et al., (2009)*. Promoter constructs (500 ng), *Myocd* (250 ng) or control *LacZ* expression plasmid and 50 ng of Renilla were transfected into Cos cells with Lipofectamine 2000 at 50% confluency. Luciferase intensity was analyzed 20 hr post-transfection.

Firefly and Renilla luciferase activities were quantified in lysates with the Dual Luciferase Reporter Assay kit (Promega) on a Victor 1420 Multilabel Counter (PerkinElmer, Madison, WI). Firefly luciferase values were normalized to Renilla to control for transfection efficiency.

## Quantitative real-time PCR

RNA was isolated with TRIzol reagent according to the manufacturer's protocol. Quantitative real-time PCR for microRNAs was performed with the TaqMan miRNA assay kit (Applied Biosystems/Life), and TaqMan probes for *miR-1* (Applied Biosystems/LIfe), *miR-133a* (Applied Biosystems/Life) were used according to the manufacturer's protocols. cDNA for mRNA quantification was generated using Superscript with oligodT and random hexamers (Invitrogen). Detection of splice variants for *Myocd* was performed using oligo dT generated cDNA (Invitrogen/Life). qPCR probes for the *Telokin* promoter, and *Myocd* splice variants were designed using Primerquest software and synthesized by Integrated DNA Technology (IDT, San Jose, CA). Intergenic region primers were designed based on sequence published previously. The probe sequences and part numbers are listed in *Supplementary file 1*. Expression values were normalized to the expression of *U6* (Applied Biosystems/Life) for miRNA analysis or *Gapdh* (Applied Biosystems/Life) for mRNA quantification, and fold change was determined using the ΔΔCT method with SDS RQ Manager software (Applied Biosystems/Life).

## RNA sequencing

Whole hearts from E18 wild-type and *miR-1* double-knockout animals were isolated and total RNA was extracted with TRIzol (Invitrogen/Life), following the manufacturer's suggested protocol. Genomic DNA was removed using a gDNA eliminator column (Qiagen, Hilden, Germany). RNA from three hearts of each genotype was pooled, and library preparation and sequencing were performed by the Beijing Genomics Institute (BGI). In brief, polyA transcripts were enriched and paired-end reads were sequenced on a High-seq 2000 (Illumina, San Diego, CA). Reads were mapped to the mm 9 genome, Ensembl v 59 annotation, with TopHat2 (*Kim et al., 2013*). Rank expectation (*Thomas et al., 2011*) was used to identify genes that were differentially expressed between the two backgrounds, using a false discovery rate threshold of 0.1.

## Array methods

Neonatal hearts (P2) from wildtype, *miR-1-1* null and *miR-1-2$^{+/-}$:miR-1-1$^{+/-}$* double heterozygote animals were isolated and total RNA was extracted with TRIzol (Invitrogen/Life), following the manufacturer's suggested protocol. Genomic DNA was removed using a gDNA Eliminator column (Qiagen). The samples were hybridized to Affymetrix Mouse Genechip ST 2.0 arrays. All arrays were RMA normalized and differentially expressed genes were identified using Limma. Gene changes were compared to an existing data set (*Zhao et al., 2007*) for *miR-1-2* null animals. 822 genes were statistically changed in one of the three genotypes when compared to the wild-type control (p value of 0.0025 and a fold change of greater than 1.5 or less than 0.5). Of these differentially expressed genes, 201 genes could be evaluated across all genotypes due to the difference in array probe sets.

## Western blots

Hearts were isolated from animals of the indicated genotypes between P0 and P3 and rinsed with 1X PBS. The tissue was resuspended in RIPA buffer and disassociated in a Bullet Blender (Next Advance). After a clarification and sonication step, the lysates were loaded on to a 4–20% SDS-PAGE (Biorad, Hercules, CA) gel and blotted using standard protocols. Primary antibodies against GAPDH (1:1000; Abcam), SRF (1:200; Santa Cruz Bio., Dallas, TX), CSRP1 (1:500; Abcam), TAGLN/Sm22 (1:500; Abcam), p-MLC ser18/thr19 (1:1000; Cell Signaling), MLCK/Telokin (1:1000; Abcam, Cambridge, England), ACTA2 (1:2000; Sigma). Visualization and quantification of blots was done on a Licor Odyssey system with fluorescently conjugated secondary antibodies (Licor, Lincoln, NE), according to manufacturer's instructions.

## Polymerase II chromatin immunoprecipation

Pol II ChIP was performed as in *Lee et al. (2006)* with minor modifications. qPCR primer sequences described in *Supplementary file 1*.

## Bioinformatics and statistical analysis

Data were analyzed with Prism and/or Excel and an unpaired *t* test was used to determine statistical significance. Predicted genotype ratios were calculated by chi-square analysis. Relative quantity for qPCR analysis was determined using ΔΔCT method. Values for experimental animals normalized to the average of controls.

## Acknowledgements

The authors thank G Howard and AL Lucido for their editorial comments; B Taylor for help with manuscript and figure preparation; K Cordes Metzler, I King, V Vedantham and M White for critical discussion; A Williams, S Thomas, and A Holloway of the Gladstone Bioinformatics Core, J Wong of the Gladstone Electron Micropscopy Core, and C Miller and K Bummer of the Gladstone Histology Core for technical support. The *Telokin* promoter construct was a gift from P Herring. RNA from *miR-133a* knockout animals was kindly provided by N Liu and EN Olson. The Gladstone Institutes received support from a National Center for Research Resources Grant RR18928.

## Additional information

### Competing interests

DS: Member of the Scientific Advisory Boards of iPierian and RegeneRx Biopharmaceuticals, and Reviewing editor, *eLife*. The other authors declare that no competing interests exist.

### Funding

| Funder | Grant reference number | Author |
|---|---|---|
| National Science Foundation | | Amy Heidersbach |
| National Institutes of Health | U01 HL100406, U01 HL098179, R01 HL057181, P01 HL089707 | Deepak Srivastava |
| American Heart Association | | Yen-Sin Ang, Kathryn N Ivey |
| Amyotrophic Lateral Sclerosis Association | | Kathryn N Ivey |
| California Institute for Regenerative Medicine | | Deepak Srivastava |
| William Younger Family Foundation | | Deepak Srivastava |
| LK Whittier Foundation | | Deepak Srivastava |
| Roddenberry Foundation | | Deepak Srivastava |

The funders had no role in study design, data collection and interpretation, or the decision to submit the work for publication.

### Author contributions

AH, KNI, Conception and design, Acquisition of data, Analysis and interpretation of data, Drafting or revising the article; CS, KC-M, YH, Acquisition of data, Drafting or revising the article; Y-SA, Analysis and interpretation of data, Drafting or revising the article; DS, Conception and design, Analysis and interpretation of data, Drafting or revising the article; PDJ, Made the *miR-1-1* targeting vector used in this study

### Ethics

Animal experimentation: This study was performed in strict accordance with the recommendations in the Guide for the Care and Use of Laboratory Animals of the National Institutes of Health. All of the animals were handled according to approved institutional animal care and use committee (IACUC) protocols (#AN086606) of the University of California, San Francisco.

## Additional files

### Supplementary files

• Supplementary file 1. Probe sequence/part number.

## Major dataset

The following datasets were generated:

| Author(s) | Year | Dataset title | Dataset ID and/or URL | Database, license, and accessibility information |
|---|---|---|---|---|
| Heidersbach A, Saxby C, Carver-Moore K, Huang Y, Ang Y-S, Ivey KN, Srivastava D | 2013 | microRNA-1 regulates sarcomere formation and suppresses smooth muscle gene expression in the mammalian heart | SRP029956; http://www.ncbi.nlm.nih.gov/sra/?term=SRP029956 | Sequence Read Archive (SRA). |
| Heidersbach A, Saxby C, Carver-Moore K, Huang Y, Ang Y-S, Ivey KN, Srivastava D | 2013 | Wildtype, miR-1-1 KO, miR-1 Double het P2 mixed strain heart analysis (MoGene 2.0 ST Arrays) | GSE51394; http://www.ncbi.nlm.nih.gov/geo/query/acc.cgi?acc=GSE51394 | Gene Expression Omnibus (GEO). |

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
