## [Decision Letter]

Thank you for sending your work entitled “microRNA-1 regulates sarcomere formation and suppresses smooth muscle gene expression in the mammalian heart” for consideration at *eLife*. Your article has been favorably evaluated by a Senior editor, a Reviewing editor, and 3 reviewers.

The Reviewing editor and the reviewers discussed their comments before we reached this decision, and the Reviewing editor has assembled the following comments to help you prepare a revised submission.

This manuscript describes the effect of compound deletion of *miR-1-1* and *miR-1-2*. The authors had previously published on the phenotype of *miR-1-2* deletion and they now show the phenotype of *miR-1-1* and double *miR-1-1/2* mutants. Complete deletion of *miR-1* results in death and cardiac dysfunction. The authors have uncovered an interesting mechanism to explain the phenotype. They show that *miR-1*-null cardiomyocytes have abnormal sarcomere organization and decreased phosphorylation of the regulatory myosin light chain-2 (MLC2). The smooth muscle restricted inhibitor of MLC2 phosphorylation, Telokin, is ectopically expressed in the myocardium, along with other smooth-muscle genes. The authors determine that the mechanism underlying *miR-1* repression of Telokin expression is both by direct targeting via seed sites in the 3' UTR and also by repressing the transcriptional regulator, Myocardin. These data therefore imply that *miR-1* functions to regulate the maintenance of the striated muscle phenotype in cardiomyocytes. This is an interesting study. However the following points need to be addressed before publication.

1) The authors claim that *miR-1-1* knockout mice have a comparable phenotype to the *miR-1-2* knockout animals and that deletion of one of the isoforms is responsible for roughly half the total amount of *miR-1* present in the heart. What are the targets that are responsible for the phenotype in the *miR-1-1* knockout mice and do they overlap with the targets that the authors previously identified to be responsible for the *miR-1-2* knockout phenotype? Are the *miR-1* targets described in Figure 4 also regulated in either the *miR-1-1* or *miR-1-2* knockout mice?

2) What are the differences between the *miR-1-2* het/*miR-1-1* het and the phenotype of either the *miR-1-1* or *1-2* homozygous null?

3) *miR-133a* is also reduced in the *miR-1-1* KO based on Figure 1. How much of the double *miR-1* mutant phenotype is due to loss of *miR-133*? This question is considered indirectly in a few instances but not really adequately addressed. The authors have RNA sequence data that they can use to predict *miR-1* (as shown in Figure 4) and *miR-133* target genes. If, as the authors suggest, reduced *miR-133a* is only a minor part of the phenotype they observe then there should not be an enrichment in the upregulated genes for *miR-133a* seed sites. If *miR-133a* regulated genes are also important for the phenotype then this can be further validated and incorporated into the story.

4) If Myocardin is upregulated in the *miR-1* knockout mice, would one not expect to see an increase in *miR-133* instead of a decrease?

5) Myocardin upregulation seen on the RNAseq dataset should show the different isoforms. If so this should be included in the paper and if not then this should be explained/discussed.

6) Regarding the RNAseq dataset please comment on why there are equal numbers of up and down regulated genes – perhaps the stage that was analyzed is too late since there would be predicted to be more genes upregulated.

7) Due to the artificial nature of luciferase experiments, one could envision that Telokin is solely regulated by *miR-1* at the transcriptional level. Can the authors comment on this?

8) In general, the phenotypic analysis is strong. The authors may consider showing some of the ECG data with different types of atrial arrhythmias.

9) The Results section can be written more concisely with part of the text moved to the Discussion.

10) Did the authors check to see whether there is a phenotype in the skeletal muscles of the compound *miR-1* knockout mice? This is not essential for this paper, but a mention in the Discussion would be interesting.

---

## [Author Response]

*1) The authors claim that* miR-1-1 *knockout mice have a comparable phenotype to the* miR-1-2 *knockout animals and that deletion of one of the isoforms is responsible for roughly half the total amount of* miR-1 *present in the heart. What are the targets that are responsible for the phenotype in the* miR-1-1 *knockout mice and do they overlap with the targets that the authors previously identified to be responsible for the* miR-1-2 *knockout phenotype? Are the* miR-1 *targets described in*
Figure 4
*also regulated in either the* miR-1-1 *or* miR-1-2 *knockout mice*?

The most prevalent phenotype in both the *miR-1-1* and *miR-1-2* knockout animals was a cardiac conduction abnormality, which was at least partially ascribed to dysregulation of the *miR-1* target, *Irx5*, in the *miR-1-2* knockout. We also found that *Irx5* was dysregulated in the *miR-1-1* knockout and present that data as Figure 1—figure supplement 6. The other *miR-1* target that may have contributed to the *miR-1-2* phenotype was *Hand2*, which was not dysregulated at the mRNA level in the *miR-1-1* knockout. More broadly, we observed similar shifts in gene expression between the *miR-1-1* and *miR-1-2* knockout mice (see number 2 below, Figure 2—figure supplement 2), some of which may represent direct targets. Most of the putative *miR-1* targets described in Figure 4 dysregulated in the *miR-1* double knockout mice were not significantly altered in the single knockouts.

*2) What are the differences between the* miR-1-2 *het/*miR-1-1 *het and the phenotype of either the* miR-1-1 *or* 1-2 *homozygous null*?

Since the purpose of this study was to assess the effect of compete loss of *miR-1*, we intentionally presented a limited phenotypic analysis of the *miR-1-1* knockout mice and then proceeded to use them as a tool to generate *miR-1*-null animals. Nevertheless, to address this question, we performed gene expression analyses to compare the *miR-1-1* and *miR-1-2* knockouts to the double heterozygotes. We have included these results as Figure 2—figure supplement 2. In summary, we find that gene expression changes of each single knockout were quite similar though not identical to that of the double heterozygotes.

*3)* miR-133a *is also reduced in the* miR-1-1 *KO based on*
Figure 1*. How much of the double* miR-1 *mutant phenotype is due to loss of* miR-133*? This question is considered indirectly in a few instances but not really adequately addressed. The authors have RNA sequence data that they can use to predict* miR-1 *(as shown in*
Figure 4*) and* miR-133 *target genes. If, as the authors suggest, reduced* miR-133a *is only a minor part of the phenotype they observe then there should not be an enrichment in the upregulated genes for* miR-133a *seed sites. If* miR-133a *regulated genes are also important for the phenotype then this can be further validated and incorporated into the story*.

Thank you for this suggestion, which we agree is a very important one. To address this comment, we utilized the GREAT interface to evaluate the enrichment of putative miRNA targets within the set of genes that were upregulated in the *miR-1*-null hearts compared to wild-type controls. We found that genes containing *miR-1/206* seed sequence complementarity were most significantly enriched in this dataset. Genes targeted by *miR-495*, *miR-518a-2*, *miR-501* and *miR-409* were also enriched, though to a lesser degree. Notably, *miR-133a* predicted targets were not found to be upregulated in the *miR-1* null hearts, providing confidence that reduced *miR-133a* is only a minor part of the phenotype and may not reach the threshold for functional significance. These data have been added to the Results and are presented as Figure 4—figure supplement 1.

*4) If Myocardin is upregulated in the* miR-1 *knockout mice, would one not expect to see an increase in* miR-133 *instead of a decrease*?

If Myocardin were the only determinant of *miR-133* expression, then we would indeed expect to see an increase in *miR-133* in the *miR-1* knockout mice. However, transcriptional regulation of the *miR-1/133* loci are coordinately regulated by several transcription factors including Myocd, SRF, Mef2, Myod, MyoG and Nkx2-5. Additionally, our data show that the increase in Myocardin expression in *miR-1* knockout mice is primarily due to aberrant expression of the smooth muscle Myocardin isoform; thus we would not necessarily expect to see upregulation of transcription of any one striated muscle-enriched Myocardin target.

*5) Myocardin upregulation seen on the RNAseq dataset should show the different isoforms. If so this should be included in the paper and if not then this should be explained/discussed*.

The smooth muscle isoform of Myocardin is distinguished by inclusion of Exon 2A, which is comprised of only 44 nucleotides. This fact alone makes the difference difficult to observe in an RNAseq dataset. Additionally, as a transcription factor, Myocardin is expressed at very low levels and the smooth muscle isoform comprises only a relatively small subset of the total Myocardin transcript in cardiomyocytes. Therefore, at the depth of our RNA-seq analysis, the read coverage at that Exon was limiting and precluded detection of its differential expression with any statistical confidence. We have added a sentence to the Results to address this apparent discrepancy.

*6) Regarding the RNAseq dataset please comment on why there are equal numbers of up and down regulated genes* – *perhaps the stage that was analyzed is too late since there would be predicted to be more genes upregulated*.

Although miRNAs are known to be “repressors”, we would not have predicted that there would be a predominance of upregulated genes in our RNAseq dataset because 1) dysregulation of transcriptional regulators or members of signaling pathways will undoubtedly cause both upregulation and downregulation of numerous downstream genes; and 2) miRNAs often repress translation rather than promote degradation of their target mRNAs. We would, however, expect to have an enrichment of mRNAs with a *miR-1* binding site in the upregulated set compared to the downregulated set of genes, and this indeed was the case (see point 3). To clarify this issue, we added a sentence to the Results section stating:

“Given that many direct miRNA targets are upregulated at the protein, but not transcript level, we expected that sequencing analysis of this stage would identify pathways that are dysregulated in the *miR-1* knockout, some of which may involve direct *miR-1* targets.”

*7) Due to the artificial nature of luciferase experiments, one could envision that Telokin is solely regulated by miR-1 at the transcriptional level. Can the authors comment on this*?

This point is certainly valid and deserving of comment in the text. We have added a sentence to the Discussion stating:

“Although it is not possible to distinguish the contribution of transcriptional regulation of Telokin vs direct *miR-1* targeting of the Telokin 3’ UTR, the dramatic degree of Telokin protein upregulation compared to the more moderate *Telokin* mRNA increase is consistent with some degree of translational control by *miR-1* (Figure 4).”

*8) In general, the phenotypic analysis is strong. The authors may consider showing some of the ECG data with different types of atrial arrhythmias*.

We thank the reviewers for this suggestion and in response we have added Figure 1—figure supplement 5 and Figure 2—figure supplement 4 showing representative ECG tracings.

*9) The Results section can be written more concisely with part of the text moved to the Discussion*.

We have edited the Results to be more concise and moved text to the Discussion, as appropriate.

*10) Did the authors check to see whether there is a phenotype in the skeletal muscles of the compound miR-1 knockout mice? This is not essential for this paper, but a mention in the Discussion would be interesting*.

Electron microscopy analysis of P0 *miR-1*-null skeletal muscle showed the presence of ordered sarcomeric structures. The newborn mice were mobile and could feed, but because of their early demise, we could not assess their muscle strength more rigorously, particularly under stress. *miR-206* continues to be expressed in skeletal muscle so a triple knockout of *miR-1-1, miR-1-2* and *miR-206* will be required to assess the consequences of complete loss of the *miR-1/miR-206* family in skeletal muscle. We have added a paragraph to the Discussion addressing this interesting finding and its possible interpretations.